# Family Life Cycle, Asset Portfolio, and Commercial Health Insurance Demand in China

**DOI:** 10.3390/ijerph192416795

**Published:** 2022-12-14

**Authors:** Ling Tian, Haisong Dong

**Affiliations:** 1School of Economics and Management, Wuhan University, Wuhan 430072, China; 2National Institute of Insurance Development, Wuhan University, Ningbo 315100, China

**Keywords:** commercial health insurance demand, family life cycle, family asset portfolio, China

## Abstract

Based on the cross-sectional data of the China household finance survey (CHFS) in 2017, this paper aims to empirically examine the effects of the family life cycle, financial status, and asset portfolio on commercial health insurance demand (breadth and depth) by constructing Probit and Tobit models, respectively. Based on all of the samples, it has been found that family life cycle, family financial status, and family asset portfolio have different influences on the breadth and the depth of health insurance. In terms of the family life cycle, there is an “inverted U-shaped” relationship with the breadth and the depth of health insurance, and the effect is obvious. In terms of family financial status, total household consumption has positive and significant effects on the breadth and depth of health insurance. Total household income and total household debt only have a significant positive impact on health insurance breadth. The total household asset portfolio is only positively correlated with health insurance depth. In terms of the family asset portfolio, the share of real estate assets has a crowding out effect on the breadth and the depth of health insurance. The share of savings assets has no significant effect on the breadth and the depth of health insurance but is positively correlated with the former and negatively correlated with the latter. Both the share of vehicle assets and the share of investment assets only have a significant impact on the breadth of health insurance; however, the positive and negative correlation is different. Based on the sub-samples, it has been found that the breadth and the depth of health insurance vary greatly in the regions and household registration characteristics.

## 1. Introduction

As a basic social unit, the family plays an important role in social and economic activities [1,2]. Family asset allocation behavior not only determines family living standards [3,4], but also has an important impact on national economic development. After the implementation of the reform and opening-up policy, the commercial health insurance industry switched into the fast development lane [5]. In terms of insurance density, the per capita density of commercial health insurance increased from CNY 50.52 in 2010 to CNY 578.82 in 2020. In terms of the depth of commercial health insurance, it has increased from 0.164% in 2010 to 0.804% in 2020. In terms of the number of companies, it has grown from 1 in 1982 to 164 in 2020. In terms of premium income, it increased from CNY 11.78 million in 1985 to CNY 844.7 billion in 2021. In terms of the growth rate, during the period from 2012 to 2021, the average annual growth rate of commercial health insurance in China reached 29.72%, far exceeding the average annual growth rate of life insurance premium income of 13.59%, life insurance premium income of 11.02%, and accident insurance premium income of 13.96% in the same period. The commercial health insurance industry has become one of the fastest growing sectors of China’s national economy during the same period, and the commercial health insurance market has become the huge blue ocean market with high growth and high potential in China and even in the rest of the world.

However, as an important part of family assets, commercial health insurance still has a low popularity rate in the family, and its position in the family assets portfolio is not high. According to the China Household Wealth Survey Report (2018) [6], the per capita wealth of Chinese households was CNY 0.1943 million in 2017, and the share of real estate net worth in the household wealth was as high as 66.35%, while the share of the household financial assets in the household wealth was only 16.26%, which is much lower than the 42.6% in the United States, 61.1% in Japan, and 56.0% in Singapore. In the proportion of the household financial assets portfolio, the sum of the cash deposits, the demand deposits, and the time deposits accounts for more than 80% of all financial assets; the value of the insurance products accounts for only 17% of all financial assets, and the proportion of commercial health insurance is even more insignificant. In addition, the proportion of commercial health insurance in the urban and rural households of China and households in eastern and midwestern regions of China is also significantly different. The division of the eastern, middle, and western regions of China is shown in Figure 1. The household asset structure is single and the participation degree of the commercial health insurance market is low, which is not conducive to the preservation or the appreciation of household wealth, nor can it provide sufficient impetus for the operation of the whole national economy. Based on this fact, this paper takes CHFS2017 [7] data as research samples in order to explore the impact of the family life cycle, the family financial status, and family asset portfolio allocation on the demand for commercial health insurance, and analyzes the urban–rural heterogeneity and regions’ heterogeneity, which has important practical significance for the formulation of macroeconomic policies, the rational allocation of family assets, and the promotion of the stable development of commercial health insurance and the improvement of family well-being in China and even in other developing countries.

The innovations of this paper are as follows: Firstly, the existing literature has not yet studied the impact of family asset portfolio allocation on commercial health insurance demand. This paper makes the first attempt to empirically study the impact of the family life cycle, the family financial status, and family asset portfolio allocation on commercial health insurance demand (breadth and depth of health insurance), and on this basis, to explore whether the breadth and the depth of family commercial health insurance have the same influencing factors. Secondly, according to the research conclusions, it provides a certain reference value for optimizing family asset allocation, promoting the stable development of health insurance and deeply understanding the national “three-child policy”.

The remainder of this paper is arranged as follows: the second part is the literature review and the hypothesis proposal; the third part is the research design, which introduces the selection and the explanation of the variables, the data sources that are used in the empirical part, the setting, and the estimation methods of the model; the fourth part is the empirical analysis, including a multicollinearity test and a heteroscedasticity test, an analysis of the empirical results, a robustness test, and an explanation of endogeneity; the fifth part is the further discussion, namely the heterogeneity analysis; and the last part is the conclusions, the policy implications, and the research prospects.

## 2. Literature Review and Hypothesis Proposal

The theoretical and empirical studies on the family life cycle, the family financial situation, family asset portfolio allocation, and insurance demand have important reference value for this study. In terms of the family life cycle, Yaari (1965) [8] built an uncertain life cycle model in order to determine the optimal level of insurance demand based on the utility maximization of the insured. Showers and Shotick (1994) [9] used the data of the US Consumer Expenditure Survey (CES) and found that the age of the household head, the household income, and the family size all had a significant positive impact on the demand for life insurance, and the positive impact of the household head age and the family size showed a decreasing trend. Bernheim et al. (2003) [10], based on the idea of family dynamic life cycle consumption, and combined with the expected mortality, the expected income, and the expected expenditure of both husband and wife, calculated the benchmark life insurance holdings, which can fully compensate the economic losses that are caused by the death of one spouse to the other. Wang et al. (2013) [11] used the micro data of the China Household Income Survey (CHIPS) in 2003 in order to study the selection of life insurance, savings, stocks, and real estate assets of urban households in China from two aspects of the household life cycle risk and household financial asset portfolio allocation.

In terms of the family financial status, Hammond et al. (1967) [12] used data from between 1953 and 1962 to study the relationship between life insurance expenditure and family economic status and demographic characteristics. Mantis and Farmer (1968) [13] investigated the specific factors affecting the demand for life insurance from the perspective of the family economic status. Anderson and Nevin (1975) [14] took newly married couples as the research object, analyzed the characteristics of their insurance consumption, and found that too high or too low current and future expected income would lead to higher-than-average family demand for life insurance. Guiso and Jappelli (1998) [15] found that families with higher income volatility had a higher demand for accident insurance. Hau (2000) [16] believed that the net assets and wealth are positively correlated with the life insurance demand. Albouy and Blagoutine (2001) [17] found that families with more income or asset accumulation have a higher intention to purchase commercial insurance, that is, the family economic status is an important factor affecting family commercial insurance decisions. Liu et al. (2003) [18] made use of the data of the China Health and Nutrition Survey (CHNS) and found that income had an impact on the insurance demand, but the relationship between low income and the insurance demand was more significant. Chang et al. (2004) [19] believed that the number of children, the time span for growing up to be independent, and the living standard of the beneficiaries were the main factors affecting the demand for life insurance. Mocan et al. (2004) [20] used the survey data of 6407 urban households in 70 cities in 10 provinces of China in order to divide medical expenditure into two sub-processes of determining whether the expenditure is incurred and the amount of medical expenditure. A two-part model and a discrete factor model were established, respectively. The determining factors of the medical demand of the rural family in China were studied. It was found that income had an effect on the medical demand, and the elasticity of income was 0.3. Lin and Grace (2007) [21] constructed a comprehensive index to measure the family financial risk. Using the data of the American consumer financial survey, the empirical study found that the family financial risk index had a significant impact on both the family-term life insurance holdings and the overall life insurance holdings. Calvett and Sodini (2014) [22] found a significant positive correlation between income and the participation in risky financial markets. Christiansen et al. (2015) [23] believed that the income and wealth level of married families and the effectiveness of increasing the couple’s joint decision making provide certain protection for all kinds of risks that are faced by families. Shi et al. (2015) [24] built Probit and Tobit models that were based on the data of the China Household Income Survey (CHIPS) in 2002. The empirical study concluded that the current wealth and the future income of households both have a curved impact on the demand for life insurance. Saavedra (2017) [25] built a Probit model that was based on the data of the US Current Population Survey (CPS) from 2001 to 2013. The empirical results showed that the total family income was significantly positively correlated with the demand for children’s private health insurance and was significantly negatively correlated with the demand for children’s public health insurance. Xiao (2018) [26], by mining the Chinese household finance survey data (CHFS) and using the Probit model and Tobit model for an empirical test, came to the conclusion that the total household assets, the household wealth level, and the household consumption expenditure all had a significant impact on the demand for commercial health insurance, and there was age and gender heterogeneity. Wang et al. (2018) [27] collected the survey data of 1842 households in the Qinghai and Zhejiang provinces, obtained the demand for long-term care insurance (LTCI) based on the conditional valuation method, and analyzed the relevant factors of the demand for long-term care insurance (LTCI) by using logistic regression with random effects. The study found that household income was significantly related to the demand for long-term care insurance (LTCI). Segodi and Sibindi (2022) [28] built a panel data model by using the data of a panel of the BRICS bloc of countries from 1999 to 2020, and the empirical results showed that income had a significant negative effect on the demand for life insurance.

In terms of family asset portfolio allocation, Headen and Lee (1974) [29] believed that household financial asset portfolio behavior had a significant impact on insurance consumption demand. Through theoretical analysis, Meyer and Ormiston (1995) [30] believed that family insurance holding decisions were an important part of family financial asset decision making, and family financial asset portfolio behavior was a key factor affecting family insurance demand. Giesbert et al. (2011) [31] used the survey data of 350 families in central Ghana in 2008 in order to build a multivariate Probit model and empirically studied the relationship between other family financial assets and the demand for life insurance. Based on the above research results, this paper proposes the following hypotheses:

**Hypothesis 1.** *The family life cycle, family financial status, and family portfolio allocation all have an impact on the demand for commercial health insurance (the breadth and depth of health insurance), but the impact on the breadth and the depth of commercial health insurance is not the same*.

**Hypothesis 2.** *The impact of the family life cycle, family financial status, and family asset portfolio allocation on the demand for family commercial health insurance (the breadth and depth of health insurance) in different household registration types and regions is different*.

## 3. Study Design

### 3.1. Variable Specifications

#### 3.1.1. Explained Variables

The breadth of commercial health insurance is defined as follows: The “whether family members buy commercial health insurance” in the questionnaire is taken as the quantitative indicator of the breadth of commercial health insurance, which is a dummy variable. If at least one family member buys commercial health insurance, it is assigned a value of 1; otherwise, it is 0, and it is taken as the explained variable of the Probit model in this paper.

The depth of commercial health insurance is defined as follows: the “natural logarithm of total premium expenditure of family members buying commercial health insurance last year” in the questionnaire is taken as a quantitative indicator of the depth of commercial health insurance, and it is taken as the explained variable of the Tobit model in this paper.

#### 3.1.2. Core Explanatory Variables

The family life cycle is defined as follows: referring to the practices of Poterba and Samwick (2001) [32] and Shum and Faig (2005) [33], “the actual age of the head of the household” and “the square of the actual age of the head of the household” in the questionnaire are taken as quantitative indicators of the family life cycle.

The family financial status is defined as follows: the natural logarithms of the “total household income”, the “total household consumption”, the “total household assets”, and the “total household debt” in the questionnaire are respectively processed as quantitative indicators of the family financial status.

The family asset portfolio is defined as follows: The share of household real estate assets, the share of household vehicle assets, the share of household savings assets, and the share of household investment assets are taken as the quantitative index of household asset portfolio. Among them, the household savings assets have low risk and low profitability, therefore this paper uses the sum of “cash”, “demand deposit”, “time deposit”, “gold value”, and “bond market value” in the questionnaire. The household investment assets have high risks and high returns; therefore, this paper uses the sum of the “stock market value”, the “financial derivatives market value”, the “fund market value”, and the “financial product market value” in the questionnaire. 

#### 3.1.3. Control Variables

In order to minimize the impact of model endogeneity on the estimation bias of this study, the following control variables are added: urban household registration or not, dummy variables (if yes, its value is 1; if no, its value is 0); eastern region or not, dummy variable (if yes, its value is 1; if no, its value is 0); the health index [34], in which the “self-rated health status of the household head” in the questionnaire is taken as the quantitative index of the health index, and it is numerically processed and assigned the values of 1–5 from “very good” to “very poor”, in which “very good” = 1, “good” = 2, “average” = 3, “poor” = 4, and “very poor” = 5.

The name, the unit of measurement, the symbol, and the definition of each variable are shown in Table 1.

### 3.2. Data Introduction

The data in this paper are from the China household finance survey (CHFS), which was conducted nationwide by Southwestern University of Finance and Economics in 2017. The data sample covers 29 provinces (excluding Xinjiang, Tibet, Hong Kong, Macao, and Taiwan), 355 districts and counties, and 1428 village (residential) committees in China, covering the assets, the liabilities, the income and the expenditure, the social security, the insurance, and the subjective attitudes of 40,011 households, and providing a comprehensive and detailed description of family economic and financial behaviors. It lays a solid foundation for the smooth development of this paper’s research work. In this paper, the data preprocessing software is “Excel”, and the econometric analysis software is “Stata”. In order to minimize the impact of heteroscedasticity on the empirical model, natural logarithms are adopted for all of the variables with currency as the unit of measurement. In order to minimize the influence of extreme values on the research results, all of the data of the continuous variables are winsorized based on 5% and 95% quantiles. In order to maintain the rationality of the sample data, this paper screens the samples of the families whose heads are between 25 and 70 years old and deletes the sample of households with missing information of the total household income, the total household debt, the total household assets, or the total household consumption. Finally, the total number of valid samples in this paper is 16,919.

See Table 2 and Table 3 for specific descriptive statistics.

Table 2 compares the differences in the family life cycle, the financial status, and the asset portfolio between the households with and those without health insurance. From the perspective of the family life cycle, the average age of the head of the family with health insurance is significantly lower than that of the family without health insurance (49 years old and 55 years old, respectivley). It shows that the life cycle of the families who purchase health insurance and those who do not purchase health insurance are, respectively, in the middle age family and the old age family. From the perspective of the family financial status, the mean of the total assets, the total income, the total consumption, and the total liabilities of the households that purchase health insurance are significantly higher than those of the households that do not purchase health insurance (14.22 > 13.58; 15.43 > 15.41; 11.29 > 10.76; and 6.04 > 4.89, respectivley), and the standard deviations of the total assets and the total consumption of the households that purchase health insurance are significantly lower than those that do not purchase health insurance (0.911 < 0.917; 0.638 < 0.768, respectively), indicating that the financial status of the households that purchase health insurance is generally good and stable. From the perspective of the family asset portfolio, the mean of share of the real estate assets, the vehicle assets, the investment assets, and the savings assets of the households that purchase health insurance is higher than that of the households that do not purchase health insurance (53.26 > 42.86; 26.77 > 15.11; 1.16 > 0.41; and 4.34 > 3.28, respectively), indicating that the asset portfolio allocation of the households that purchase health insurance is more optimized and diversified.

Table 3 shows the descriptive statistics of the main variables. In terms of the standard deviation index, of all of the variables, except for head_age, head_age2, house_asset, and vehicle_asset (9.940; 1042.743; 26.894; and 18.468, respectively), the random fluctuation range of the other variables is small and basically stays between 0 and 5.5, indicating that these variables have stability characteristics. In terms of the skewness and kurtosis index, except for variables H, lntotal_inc, saving_asset, and invest_asset, the other variables are generally close to normal distribution, among which variables H, lntotal_inc, saving_asset, and invest_asset have right-skewed distribution (2.851 > 0; 1.962 > 0; 1.690 > 0; and 2.901 > 0, respectively) with sharp peaks and steep features (9.129 > 3; 4.848 > 3; 4.857 > 3; and 10.002 > 3, respectively).

### 3.3. Model Setting

In order to more accurately examine the impact of the family life cycle, the financial status, and the asset portfolio on the consumption behavior of commercial health insurance, this paper draws on the practices of Cragg (1971) [35], Duan et al. (1983) [36], Pohlmeier and Ulrich (1995) [37], Cardik and Wilkins (2009) [38], Shi et al. (2015) [24], and Xiao (2018) [26],. First of all, the Probit model is adopted in order to analyze the influence of the family life cycle, the financial status, and the asset portfolio on the breadth of commercial health insurance, and then the left truncated Tobit model is constructed in order to analyze the influence of the family life cycle, the financial status, and the asset portfolio on the depth of commercial health insurance. The expression of the Probit model is as follows:(1)$$\{\begin{array}{l} H=I\;(H^{*}>0)\\ H^{*}=\alpha +\beta \cdot Core\_Variable+\gamma \cdot Control\_Variable+\varepsilon \end{array}$$
where *H* represents whether a family holds commercial health insurance, which is a dummy variable, namely the breadth of family commercial health insurance, and *I* (.) is an indicative function. If the expression in parentheses is true, the value is 1; otherwise, the value is 0; *H** represents potential variables that cannot be observed; *Core_Variable* represents all of the core independent variables; *Control_Variable* represents all of the control variables; *ε* is a random perturbation term and follows the standard normal distribution.

However, in the analysis of family commercial health insurance depth, only the premiums that are paid by the families with commercial health insurance can be observed (the value is positive), while the premiums that are paid by the families without commercial health insurance cannot be observed, that is, the depth of family commercial health insurance is truncated. Therefore, the Tobit model is used in this paper for estimation and analysis. The model expression is as follows:(2)$$\{\begin{array}{l} \ln H\_expense^{*}=\phi +\tau \cdot Core\_Variable+\lambda \cdot Control\_Variable+v\\ \ln H\_expense=\max(0,\ln H\_expense^{*}) \end{array}$$
where ln*H*_*expense* represents the natural logarithm of the family’s actual commercial health insurance premium expenditure last year, namely, the depth of the family commercial health insurance; ln*H_expense** indicates the observed value if ln*H_expense* > 0. *Core_Variable* represents all of the core independent variables; *Control_Variable* represents all of the control variables; and v is a random perturbation, which follows a normal distribution

## 4. Empirical Analysis

### 4.1. Multicollinearity Test and Heteroscedasticity Test

Before the empirical analysis, the multicollinearity test and the heteroscedasticity test are conducted (see Table 4 and Table 5). As shown in Table 4, the variance enlargement factor (VIF) of all of the core independent variables is less than 10, indicating that Model 1 and Model 2 do not have serious multicollinearity problems. As shown in Table 5, the Chi-square statistical values of the White test for Model 1 and Model 2 are 1658.45 and 159.97, respectively, and the *p* values are 0.0000 and 0.0001, respectively, both of which have passed the 1% significance level test, indicating that Model 1 and Model 2 both have the problem of heterovariance; therefore, the robust standard error is adopted in this paper in order to correct this.

### 4.2. Analysis of Empirical Results

Table 6 Column (1) and Column (2) are the regression results of Model 1 (Probit_1) and Model 2 (Tobit_1), respectively, which are also the benchmark empirical results of this paper. Since the regression coefficients of the two models themselves have little economic significance, this paper lists the average marginal effect of more practical significance. Based on this, Hypothesis 1 is verified.

From Table 6 Column (1), it can be seen that, in terms of the family life cycle, the head_age and head_age2 have significant influences on the breadth of commercial health insurance at the level of 1%. The former is positively correlated with it, while the latter is negatively correlated with it, that is, there is an “inverted U-shaped” relationship between the family life cycle and the breadth of commercial health insurance. The possible reason for this is that, with the increase in the age of the household head, the family income also keeps increasing, the wealth keeps accumulating, and the breadth of commercial health insurance also increases. However, after the age turning point, the participation in commercial health insurance will decrease due to the sharp decline in income sources and the restriction of age and health status that is set by the health insurance company on the insurance products. In terms of the family financial status, the total household income (lntotal_inc), the total household consumption (lntotal_consump), and the total household debt (lntotal _debt) all have a significant positive effect on the breadth of commercial health insurance; the former passes the significance level test of 5%, while the latter two are significant at the 1% level. There is no significant negative correlation between the total household assets and the breadth of commercial health insurance. The possible reason for this is that the increase in the total household income and the total consumption indicates that the family has a strong ability to pay for the insurance, therefore it will increase its participation in commercial health insurance. The increase in the total household debt will increase the participation in commercial health insurance for the purpose of avoiding debt crisis, spreading risks and health protection. Although the increase in the total household assets means that households have stronger self-insurance ability, and the increase in the household assets stock generally does not cause health risks, the effect is not strong enough to affect the positive effect of buying commercial health insurance; therefore, it will reduce the degree of dependence on commercial health insurance, but it has little effect. In terms of the family asset portfolio, both the real estate assets’ share (house_asset) and vehicle assets’ share (vehicle_asset) have a significant negative effect on the breadth of commercial health insurance and pass the 1% significance level test. The share of savings assets (saving_asset) has a negative effect on the breadth of commercial health insurance, but it is not significant. The influence of the investment assets’ share (invest_asset) on the breadth of commercial health insurance is significantly positive at the level of 1%. The possible reason for this is that real estate assets and vehicle assets themselves have a strong risk protection effect, therefore they will reduce the participation in commercial health insurance. With the increase in savings assets, the family’s economic situation can be protected, therefore it is not interested in the positive effect of risk protection that is brought by commercial health insurance products. Investment assets and commercial health insurance have strong complementarity, which is of great significance for families in order to build a diversified investment portfolio, therefore it will increase the participation in commercial health insurance.

From Table 6 Column (2), it can be seen that, in terms of the family life cycle, the head_age and head_age2 have a significant influence on the depth of commercial health insurance, and the relationship between them is also “inverted U-shaped”. The possible reason for this is that middle-aged families have to support both minor children and parents, and their burden is much heavier than that of young and old families. Commercial health insurance can provide economic security for families, therefore middle-aged families pay more for commercial health insurance premiums. In terms of the family financial status, both the total household assets (lntotal_asset) and the total household consumption (lntotal_consump) have a significant positive effect on the depth of commercial health insurance, and both pass the significance level test of 5%. The total household income (lntotal_inc) and the total household debt (lntotal_debt) are not significantly positively correlated with the depth of commercial health insurance. The possible reason for this is that the average cost of participating in the commercial health insurance market is lower for families with high total consumption and total assets. For them, commercial health insurance can not only provide risk protection, but also maintain and increase value. Therefore, if such families have the desire to buy commercial health insurance, they usually spend more on the premiums of commercial health insurance. For the families with high total household debt, although they are willing to buy commercial health insurance products for the purpose of risk diversification, due to the financial constraints, the positive impact on commercial health insurance premium expenditure is not significant. Families with a higher total income have a good economic foundation and can afford to buy commercial health insurance, but they can also rely on their own financial resources to deal with health risks. Therefore, they are not very willing to spend commercial health insurance premiums, and the positive effect is not obvious. In terms of the family asset portfolio, both the savings assets’ share (saving_asset) and the vehicle assets’ share (vehicle_asset) have an insignificant negative impact on the depth of commercial health insurance. The real estate assets’ share (house_asset) has a significant “crowding out effect” on the depth of commercial health insurance. There is no significant positive correlation between the investment assets’ share (invest_asset) and the depth of commercial health insurance. The possible reason for this is that the increase in the savings assets and the vehicle assets can be used to disperse risks, but the impact is not strong enough to affect the positive effect that is brought by the purchase of commercial health insurance, therefore it will reduce the premium expenditure of commercial health insurance, but it has little effect. The market value of real estate assets is high, which can bring good income protection for families, therefore it will reduce the premium expenditure of commercial health insurance. Families with high investment assets may participate in commercial health insurance because they consider investment risks and may not be able to cope with the potential health risks. However, due to the high return on investment assets, they will not take out commercial health insurance products with high premiums.

### 4.3. Robustness Test

Through the above analysis, Column (1) and Column (2) in Table 6 are the benchmark regression results of this paper. In order to ensure the reliability of the empirical conclusions of the benchmark regression model, the corresponding robustness test is carried out in this part. The robustness test results are shown in Table 7. Specifically, two methods are used in order to verify the stability of the benchmark regression conclusion. Method 1: Replace the core independent variable method. Replace the head_age and head_age2, which represent the family life cycle, with the couple_age and couple_age2. The two groups of estimation results are shown in Table 7 Column (1) and Column (3). By comparing Table 6 Column (1) and Table 7 Column (1), it can be found that the empirical conclusions of the two are basically the same. Similarly, the empirical conclusions of Table 6 Column (2) and Table 7 Column (3) are also basically the same, indicating that the estimation results of the two models (Model 1 and Model 2) are robust, that is, the baseline regression results are robust. Method 2: The model replacement method, which is based on the similarity of the applicable conditions of the model and draws on the practice of Wang et al. (2017) [39], Li and Yang (2021) [40], and Chen (2014) [41]. The Probit model of Model 1 is replaced with the Logit model, and the Tobit model of Model 2 is replaced with the Heckman model. The other two groups of estimation results are obtained as shown in Table 7 Column (2) and Column (4). By comparing Table 6 Column (1) with Table 7 Column (2), Table 6 Column (2), and Table 7 Column (4), it can be found that, compared with the baseline “Probit_1” regression and the baseline “Tobit_1” regression, the estimation coefficient, the influence direction, and the significance level of each explanatory variable have not changed generally, that is, the baseline empirical results are still robust. To sum up, the benchmark empirical conclusions of this paper have good stability.

### 4.4. Explanation of Endogeneity

In view of the possible endogeneity problems in the model, this paper has selected and controlled most of the key variables, therefore in this part, only the endogeneity problems that are caused by the possible two-way causal relationship between the core explanatory variable and the explained variable of the family financial status are considered. Therefore, in order to alleviate the regression bias that is caused by the endogenous problems, this part attempts to use the instrumental variable method (IV) for the empirical analysis. In this part, “the average financial status of other families in the same community” is selected as the instrumental variable of the family’s financial status. Namely, the “average total income of other households in the same community” (lntotal_inc1), the “average total consumption of other households in the same community” (lntotal_consump1), the “average total assets of other households in the same community” (lntotal_asset1), and the “average total debt of other households in the same community” (lntotal_debt1) are instrumental variables of the core explanatory variables lntotal_inc, lntotal_consump, lntotal_asset, and lntotal_debt, respectively. The reason why these variables are selected as instrumental variables is that they are not related to the unobstructible characteristics of the family (i.e., random disturbance term), but they can affect the financial status of the family and better satisfy the two basic conditions of instrumental variables—exogeneity and correlation.

Table 8 Columns (2) and (4) are the regression results of instrumental variable method. Table 8 Column (1) is the baseline Probit_1 regression, and Table 8 Column (3) is the baseline Tobit_1 regression for comparison with the empirical results of the instrumental variable method. The Wald test results show that the *p*-values of the family financial status of the core independent variables are 0.0135 and 0.0161, respectively. In both the breadth and the depth analysis of commercial health insurance, the family financial status of the core explanatory variable has passed the significance level test of 5%, indicating that the family financial status of the core explanatory variable is an endogenous variable, and the adoption of the instrumental variable method is correct. According to the F value of the first stage, no matter in the breadth or in the depth of commercial health insurance, the F value of all of the first stage is significantly greater than 10 (Stock and Yogo, 2005 [42]; Andrews et al., 2019 [43]), indicating that the instrumental variable of the “average financial status of other families in the same community” that is used in the paper has strong explanatory power, and there is no weak instrumental variable problem. It can be seen from Table 8 Columns (2) and (4) that, compared with the baseline regression results of Table 8 Columns (1) and (3), respectively, the regression coefficients of the core explanatory variables have slightly changed, while the influence direction and the significance of the coefficients have not changed. Therefore, it is considered that the endogeneity problem of the model has little influence on the regression results of the core independent variables, and at the same time, the robustness of the baseline regression estimation results is verified.

## 5. Further Discussion

The relationship between the family life cycle, the financial status, and the asset portfolio and the demand for commercial health insurance (breadth and depth) under the whole sample has been studied above. However, considering that there are certain differences in the family life cycle, the financial status, and the asset portfolio of the different household registration types and the different regions, the driving effect of the family life cycle, the financial status, and the asset portfolio on the demand for commercial health insurance may be heterogeneous to a certain extent. Therefore, this part refers to the practice of Murendo and Mutsonziwa (2017) [44], and divides the whole sample into urban and rural sub-samples according to the household registration types. The estimated results are shown in Table 9. Similarly, the total samples are divided into eastern and midwestern sub-samples according to the regional types, and the estimated results are shown in Table 10. Based on this, Hypothesis 2 is verified.

### 5.1. Analysis of Heterogeneity of Household Registration Types

By comparing Table 9 Column (1) and Column (3) with the baseline regression of Table 6 Column (1) and Column (2), it can be concluded that the estimation results of the urban sub-samples are similar to the results of the full sample analysis. However, for the rural sub-samples, the analysis results are obviously different from those of the full sample. The analysis results are shown in Table 9 Column (2) and Column (4). On the whole, the driving effect of the family life cycle, the financial status, and the asset portfolio of the rural households on the demand for commercial health insurance is lower than that of the urban household registrations. The reason for this may be that, compared with the urban household registrations, the life cycle of the rural households is mostly in the young and old stage, the financial status is not very good, and the awareness of using asset portfolio to disperse risks is low or even absent. Therefore, the availability of commercial health insurance to rural families is not high.

### 5.2. Analysis of Heterogeneity of Regional Type

By comparing Table 10 Column (1) and Column (3) with the baseline regression of Table 6 Column (1) and Column (2), it can be concluded that the estimation results of the sub-samples in the eastern region are similar to the results of the full sample analysis. However, for the sub-samples in the midwestern regions, the analysis results are obviously different from those of the full samples. The analysis results are shown in Table 10 Column (2) and Column (4). On the whole, the driving effect of the family life cycle, the financial status, and the asset portfolio on the demand for commercial health insurance in the central and western regions is lower than that in the eastern regions. The reason for this may be that the economy in the midwestern regions is relatively behind that of the eastern regions of China, and the health insurance companies do not carry out a lot of product business. That is, the problem of regional imbalance exists in the development of the Chinese commercial health insurance industry.

## 6. Conclusions, Implications, and Prospects

### 6.1. Conclusions

Based on the cross-sectional data of the 2017 China household finance survey (CHFS), this paper constructs Probit and Tobit models, respectively, and empirically discusses the effects of the family life cycle, the financial status, and the asset portfolio on the demand (breadth and depth) for commercial health insurance. The main research conclusions are as follows: based on all of the samples, it has been found that the family life cycle, the family financial status, and the family asset portfolio have different influences on the breadth and the depth of health insurance. In terms of the family life cycle, there is an “inverted U-shaped” relationship with the breadth and the depth of health insurance, and the effect is obvious. In terms of the family financial status, the total household consumption has positive and significant effects on the breadth and the depth of health insurance. The total household income and the total household debt only have a significant positive impact on the health insurance breadth. The total household asset is only positively correlated with the health insurance depth. In terms of the family asset portfolio, the share of real estate assets has a crowding out effect on the breadth and the depth of health insurance. The share of savings assets has no significant effect on the breadth and the depth of health insurance, but is positively correlated with the former and negatively correlated with the latter. Both the share of vehicle assets and the share of investment assets only have a significant impact on the breadth of commercial health insurance, but the positive and negative correlation is different. In terms of the sub-samples, the driving effect of the family life cycle, the financial status, and the asset portfolio of rural families and midwestern families on the demand for commercial health insurance is lower than that of urban families and eastern families, respectively, that is, the breadth and the depth of health insurance have great differences in household registration and regional characteristics.

### 6.2. Policy Implications

Based on the above main research conclusions, the policy implications of this paper are as follows:(1)Improve the age structure of the population and promote the rationalization of the family life cycle. The specific measures for this are as follows: Firstly, the government actively formulates relevant laws and regulations, such as providing maternity allowances and improving maternity insurance benefits, in order to guide young people who are of appropriate age to enter the marriage palace as early as possible, to actively publicize and implement the “open three children” policy, and to encourage newlywed families to have more children, so as to achieve a long-term balance in the population age structure. Secondly, we should actively promote the construction of a harmonious family featuring “a loving husband and wife, respecting the old, and caring for the young”, in order to promote the rationalization of the family life cycle and the accumulation of wealth, so as to create a stable micro investment unit for the development of commercial health insurance;(2)Accelerate the modernization of economic construction and improve the family financial condition. The specific measures for this are as follows: Firstly, we will appropriately raise the minimum wage and the poverty alleviation threshold, increase the living allowances for low-income families, and reduce their tax burden. Secondly, we should improve the wage increase mechanism and the payment guarantee mechanism for enterprise employees in order to raise their wage level. We will strengthen job skills training and provide more job opportunities for enterprise employees in order to further expand the scope of social employment. Thirdly, we should speed up the urbanization process of the registered population, continue to implement the strategy of developing the midwestern regions, and speed up the construction of infrastructure and supporting facilities in rural areas and the midwestern regions. Fourthly, we should take the market as a barometer, transform the traditional industries with advanced technologies in order to optimize them, and upgrade the industrial structure;(3)Optimize family asset portfolio allocation. The specific measures for this are as follows: Firstly, we will actively implement regional housing purchase restrictions and differentiated housing credit policies, do our best to stabilize the real estate market by controlling the housing prices, preventing bubbles and risks, and comprehensively standardize the real estate market. Secondly, we will strengthen the vehicle construction system, improve the vehicle management files, and improve the safe driving mechanism, in order to create a harmonious and civilized traffic atmosphere. Thirdly, we will enhance the popularization of family financial education and financial knowledge, appropriately guide more families to participate in the financial asset market, and diversify the allocation of family assets, so as to promote the steady growth of the demand for commercial health insurance.

### 6.3. Research Prospects

This paper has drawn some meaningful conclusions; however, there are still some regrets. Due to the availability of data and the limitation of the length of this paper, the mechanism of action was not tested. In view of this, on the basis of this study, the researchers can continue to collect data, select appropriate mediating variables, and conduct mediation effect analysis in the future, so as to obtain richer and more three-dimensional research results. At the same time, because there is no research on the impact of the family asset portfolio on the demand for commercial health insurance, it is necessary for us to learn from the relevant research methods in other fields for enrichment.

## Figures and Tables

**Figure 1 ijerph-19-16795-f001:**
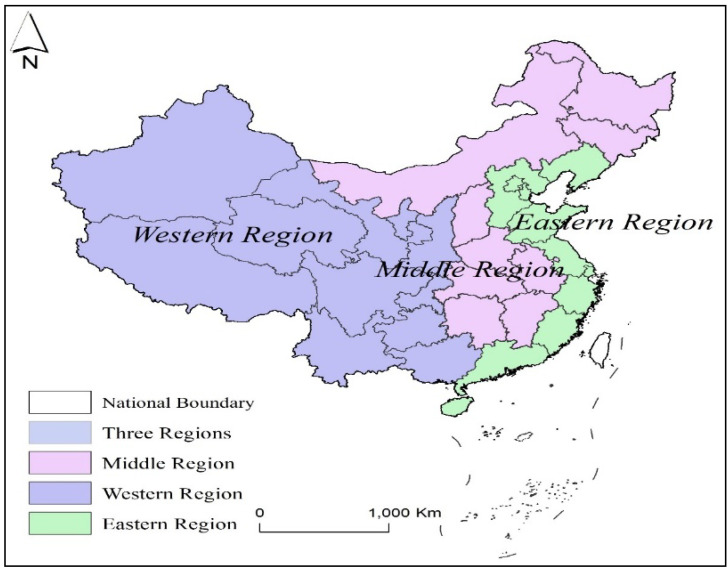
The geographical division of China.

**Table 1 ijerph-19-16795-t001:** Definition of each variable.

Name of Variable	Symbol of Variable	Explanation of Variables
Breadth of health insurance	H	Dummy variable, health insurance or not (yes = 1, no = 0)
Depth of health insurance (CNY)	lnH_expense	Natural logarithm of health insurance premium expenditure; ln (family health insurance premium expenditure + 1)
Head of household’s age (years old)	head_age	Head of household’s actual age
Square of the head of household’s age (years old)	head_age2	The square of the head of household’s actual age
Total household income (CNY)	lntotal_inc	ln (total household income + 1)
Total household consumption (CNY)	lntotal_consump	ln (total household consumption + 1)
Total household assets (CNY)	lntotal_asset	ln (total household assets + 1)
Total household debt (CNY)	lntotal_debt	ln (total household debt + 1)
Share of real estate assets (%)	house_asset	Household real estate assets/total household assets × 100
Share of vehicle assets (%)	vehicle_asset	Household vehicle assets/total household assets × 100
Share of savings assets (%)	saving_asset	Household savings assets/total household assets × 100; Household savings assets = cash + demand deposits + time deposits + gold value + bond market value
Share of investment assets (%)	invest_asset	Household investment assets/total household assets × 100; Household investment assets = stock market value + fund market value + financial derivatives market value + wealth management product value
Urban household registration or not	town	Dummy variable, urban household registration or not (yes = 1, no = 0)
Eastern region or not	east	Dummy variable, eastern region or not (yes = 1, no = 0)
Health index	health	Values from “very good” to “very poor” are 1–5; among them, “Very good” = 1, “good” = 2, “fair” = 3, “poor” = 4, and “very poor” = 5

**Table 2 ijerph-19-16795-t002:** Family life cycle, financial, and asset portfolio statistics of the sample by the possession of health insurance or not.

Variable		Families without Health Insurance			Families with Health Insurance		
		Obs	Mean	Std. Dev.	Obs	Mean	Std. Dev.
Family life cycle	head_age	15,385	54.44	9.851	1534	49.18	9.561
	head_age2	15,385	3060.66	1037.598	1534	2509.76	958.789
Family financial status	lntotal_inc	15,385	15.41	0.034	1534	15.43	0.047
	lntotal_consump	15,385	10.76	0.768	1534	11.29	0.638
	lntotal_asset	15,385	13.58	0.917	1534	14.22	0.911
	lntotal_debt	15,385	4.89	5.437	1534	6.04	5.814
Family asset portfolio	house_asset	15,385	42.86	26.981	1534	53.26	24.049
	vehicle_asset	15,385	15.11	18.592	1534	26.77	13.086
	saving_asset	15,385	3.28	4.585	1534	4.34	4.782
	invest_asset	15,385	0.41	1.270	1534	1.16	1.914

**Table 3 ijerph-19-16795-t003:** Descriptive statistics of the main variables.

Variable	Obs	Mean	Std. Dev.	Max	Min	Skewness	Kurtosis
H	16,919	0.091	0.287	1	0	2.851	9.129
lnH_expense	1534	8.251	1.216	10.12	5.3	−0.645	3.033
head_age	16,919	53.96	9.940	70	25	−0.365	2.421
head_age2	16,919	3010.72	1042.743	4900	625	−0.014	2.107
lntotal_inc	16,919	15.415	0.036	15.5	15.4	1.962	4.848
lntotal_consump	16,919	10.805	0.773	12.2	9.28	−0.098	2.339
lntotal_asset	16,919	13.638	0.935	15.6	12.4	0.565	2.332
lntotal_debt	16,919	4.991	5.483	12.9	0	0.262	1.194
house_asset	16,919	43.802	26.894	88.01	1.8	0.009	1.819
vehicle_asset	16,919	25.718	18.468	63.24	2.77	0.610	2.206
saving_asset	16,919	3.376	4.613	16.63	0.001	1.690	4.857
invest_asset	16,919	0.481	1.359	5.33	0.001	2.901	10.002

**Table 4 ijerph-19-16795-t004:** Multicollinearity test results of the core independent variables.

Core Independent Variable	Model 1			Model 2		
	The Value of VIF	The Value of 1/VIF	Result	The Value of VIF	The Value of 1/VIF	Result
head_age	2.34	0.428	Non-collinearity	2.06	0.486	Non-collinearity
lntotal_inc	1.47	0.682	Non-collinearity	1.52	0.657	Non-collinearity
lntotal_consump	1.74	0.574	Non-collinearity	1.69	0.593	Non-collinearity
lntotal_asset	8.29	0.121	Non-collinearity	7.35	0.136	Non-collinearity
lntotal_debt	1.15	0.871	Non-collinearity	1.16	0.863	Non-collinearity
house_asset	3.47	0.289	Non-collinearity	2.16	0.462	Non-collinearity
vehicle_asset	8.89	0.112	Non-collinearity	6.99	0.143	Non-collinearity
saving_asset	1.36	0.736	Non-collinearity	1.33	0.751	Non-collinearity
invest_asset	1.28	0.782	Non-collinearity	1.24	0.805	Non-collinearity

**Table 5 ijerph-19-16795-t005:** Results of the heteroscedasticity test.

Model Name	White Test		Result
	The Value of Chi-Square	The Value of *p*	
Model 1	1658.45	0.0000 ***	Heteroscedasticity
Model 2	159.97	0.0001 ***	Heteroscedasticity

Note: *** is significant at the level of 1%.

**Table 6 ijerph-19-16795-t006:** Model estimation results.

Variable	(1)	(2)
	Model 1: The Breadth of Health Insurance	Model 2: The Depth of Health Insurance
	Probit_1	Tobit_1
head_age	0.0069 *** (3.40)	0.1164 *** (4.32)
head_age2	−0.0001 *** (−4.89)	−0.0012 *** (−4.67)
lntotal_inc	0.1312 ** (2.18)	0.7616 (1.03)
lntotal_consump	0.0283 *** (7.71)	0.1273 ** (2.36)
lntotal_asset	−0.0071 (−1.17)	0.2013 ** (2.41)
lntotal_debt	0.0015 *** (3.88)	0.0003 (0.06)
house_asset	−0.0007 *** (−5.70)	−0.0041 ** (−2.40)
vehicle_asset	−0.0002 *** (−5.72)	−0.0003 (−0.49)
saving_asset	0.0002 (0.30)	−0.0028 (−0.42)
invest_asset	0.0079 *** (5.80)	0.0022 (0.14)
_cons	−17.5136 *** (−2.80)	−10.7074 (−0.97)
Control variables	Yes	Yes
Obs	16,919	16,919
Pseudo R^2^	0.1257	0.0613

Note: Except for the regression coefficient reported by _cons, the other independent variables in the table report the average marginal effect. The statistical values (Z value and *t* value, respectively) are in parentheses. *** and ** are significant at the level of 1% and 5% respectively.

**Table 7 ijerph-19-16795-t007:** Robustness test of the model.

Variable	Breadth of Health Insurance: Whether to Purchase Health Insurance		Depth of Health Insurance: Health Insurance Premium Expenditure	
	(1)	(2)	(3)	(4)
	Probit_2	Logit	Tobit_2	Heckman
head_age		0.0082 *** (3.98)		0.1175 *** (3.55)
head_age2		−0.0001 *** (−5.45)		−0.0012 *** (−3.30)
couple_age	0.0078 *** (3.48)		0.1312 *** (4.36)	
couple _age2	−0.0001 *** (−4.89)		−0.0014 *** (−4.70)	
lntotal_inc	0.1312 ** (2.19)	0.1157 ** (2.00)	0.7821 (1.06)	0.7811 (0.96)
lntotal_consump	0.0284 *** (7.74)	0.0285 *** (7.51)	0.1271 ** (2.22)	0.1335 ** (1.11)
lntotal_asset	−0.0071 (−1.18)	−0.0111 (−1.83)	0.1993 ** (2.38)	0.2007 ** (2.38)
lntotal_debt	0.0015 *** (3.89)	0.0015 *** (3.75)	0.0003 (0.05)	0.0005 (0.08)
house_asset	−0.0007 *** (−5.71)	−0.0007 *** (−5.83)	−0.0041 ** (−2.41)	−0.0042 ** (−2.11)
vehicle_asset	−0.0002 *** (−5.73)	−0.0002 *** (−6.22)	−0.0003 (−0.51)	−0.0003 (−0.36)
saving_asset	0.0001 (0.30)	0.0001 (0.17)	−0.0029 (−0.42)	−0.0027 (−0.39)
invest_asset	0.0079 *** (5.79)	0.0073 *** (5.76)	0.0022 (0.14)	0.0039 (0.12)
_cons	−17.6671 *** (−2.82)	−29.7619 *** (−2.59)	−11.3410 (−1.03)	−11.1283 (−0.84)
Control variables	Yes	Yes	Yes	Yes
Obs	16,919	16,919	16,919	16,919

Note: Except for the regression coefficient reported by _cons, the other independent variables in the table report the average marginal effect. The statistical values (Z value and *t* value, respectively) are in parentheses. *** and ** are significant at the level of 1% and 5% respectively.

**Table 8 ijerph-19-16795-t008:** Endogeneity test of the model.

Variable		Breadth of Health Insurance: Whether to Purchase Health Insurance		Depth of Health Insurance: Health Insurance Premium Expenditure	
		(1)	(2)	(3)	(4)
		Probit_1	IV-Probit	Tobit_1	IV-Tobit
head_age		0.0069 *** (3.40)	0.0479 *** (3.49)	0.1164 *** (4.32)	0.1154 *** (4.28)
head_age2		−0.0001 *** (−4.89)	−0.0007 *** (−5.03)	−0.0012 *** (−4.67)	−0.0012 *** (−4.64)
lntotal_inc		0.1312 ** (2.18)	0.9866 ** (2.05)	0.7616 (1.03)	0.4183 (0.42)
lntotal_consump		0.0283 *** (7.71)	0.1891 *** (6.90)	0.1273 ** (2.36)	0.1118 ** (1.84)
lntotal_asset		−0.0071 (−1.17)	−0.0376 (−0.86)	0.2013 ** (2.41)	0.3079 ** (3.39)
lntotal_debt		0.0015 *** (3.88)	0.0105 *** (3.79)	0.0003 (0.06)	0.0007 (0.13)
house_asset		−0.0007 *** (−5.70)	−0.0051 *** (−5.48)	−0.0041 ** (−2.40)	−0.0042 ** (−2.46)
vehicle_asset		−0.0002 *** (−5.72)	−0.0014 *** (−5.38)	−0.0003 (−0.49)	0.0003 (0.46)
saving_asset		0.0002 (0.30)	0.0015 (0.43)	−0.0028 (−0.42)	−0.0013 (−0.20)
invest_asset		0.0079 *** (5.80)	0.0531 *** (5.56)	0.0022 (0.14)	0.0002 (0.01)
_cons		−17.5136 *** (−2.8)	−18.7569 *** (−2.6)	−10.7074 (−0.97)	−6.7968 (−0.46)
Control variables		Yes	Yes	Yes	Yes
Obs		16,919	16,919	16,919	16,919
Wald test	Chi square value		12.58		12.18
	*p* value		0.0135 **		0.0161 **
F value of the first stage	lntotal_inc1		4424.04		259.32
	lntotal_consump1		41,284.97		1770.83
	lntotal_asset1		>99,999.00		5817.84
	lntotal_debt1		>99,999.00		42,157.00

Note: Except for the regression coefficient reported by _cons, the other independent variables in the table report the average marginal effect. The statistical values (Z value and *t* value, respectively) are in parentheses. *** and ** are significant at the level of 1% and 5% respectively.

**Table 9 ijerph-19-16795-t009:** Model estimation results under sub-samples divided by household registration type.

Variable	Breadth of Health Insurance: Whether to Purchase Health Insurance		Depth of Health Insurance: Health Insurance Premium Expenditure	
	(1)	(2)	(3)	(4)
	Town-Probit_1	Rural-Probit_2	Town-Tobit_1	Rural-Tobit_2
head_age	0.0117 *** (3.50)	0.0015 (0.70)	0.1023 *** (3.63)	0.2090 ** (2.57)
head_age2	−0.0002 *** (−5.05)	−0.00022 (−0.98)	−0.0011 *** (−3.92)	−0.0022 *** (−2.79)
lntotal_inc	0.1950 ** (2.06)	0.0262 (0.34)	0.8594 (1.14)	−1.3935 (−0.56)
lntotal_consump	0.0315 *** (4.60)	0.0199 *** (6.61)	0.1871 *** (3.03)	−0.0305 (−0.22)
lntotal_asset	−0.0024 (−0.23)	0.0171 ** (2.16)	0.1701 ** (1.76)	−0.0510 (−0.19)
lntotal_debt	0.0027 *** (4.18)	−0.00003 (−0.09)	0.0014 (0.26)	−0.0023 (−0.15)
house_asset	−0.0014 *** (−6.37)	0.0001 (0.90)	−0.0023 (−1.29)	−0.0137 *** (−2.77)
vehicle_asset	−0.0003 *** (−4.28)	0.00002 (0.55)	−0.0004 (−0.57)	−0.0026 ** (−1.85)
saving_asset	−0.0015 (−1.72)	0.0014 *** (3.07)	−0.0023 (−0.32)	−0.0069 (−0.38)
invest_asset	0.0115 *** (5.67)	0.0085 *** (2.87)	0.0107 (0.67)	−0.1296 (−1.51)
_cons	−17.5558 *** (−2.56)	−13.7889 (−0.86)	−11.1839 (−1.00)	26.1598 (0.69)
Control variables	Yes	Yes	Yes	Yes
Obs	8813	8106	8813	8106
Pseudo R^2^	0.0749	0.0742	0.0288	0.0223

Note: Except for the regression coefficient reported by _cons, the other independent variables in the table report the average marginal effect. The statistical values (Z value and *t* value, respectively) are in parentheses. *** and ** are significant at the level of 1% and 5% respectively.

**Table 10 ijerph-19-16795-t010:** Model estimation results under sub-samples divided by region type.

Variable	Breadth of Health Insurance: Whether to Purchase Health Insurance		Depth of Health Insurance: Health Insurance Premium Expenditure	
	(1)	(2)	(3)	(4)
	East-Probit_1	Midwest-Probit_2	East-Tobit_1	Midwest-Tobit_2
head_age	0.0056 ** (1.75)	0.0051 ** (2.02)	0.1091 *** (3.08)	0.0976 ** (2.17)
head_age2	−0.0001 *** (−3.07)	−0.00007 *** (−3.03)	−0.0013 *** (−3.65)	−0.0010 ** (−2.32)
lntotal_inc	0.0842 (0.95)	0.1468 * (1.75)	2.0327 ** (2.11)	−1.9001 (−1.51)
lntotal_consump	0.2452 *** (7.74)	0.0251 *** (5.77)	0.1448 ** (1.82)	0.1593 * (1.82)
lntotal_asset	0.0032 (0.35)	−0.0152 (−1.58)	0.1845 ** (1.72)	0.3128 * (1.90)
lntotal_debt	0.0014 *** (2.17)	0.0011 ** (2.17)	0.0027 (0.40)	−0.0050 (−0.58)
house_asset	−0.0007 *** (−3.43)	−0.0002 (−1.17)	−0.0029 (−1.20)	0.0010 ** (0.36)
vehicle_asset	−0.0002 *** (−3.57)	−0.0002 *** (−4.91)	−0.0005 (−0.57)	−0.0010 * (−1.03)
saving_asset	0.0007 (0.87)	0.0003 (0.48)	−0.0036 (−0.39)	−0.0001 (−0.01)
invest_asset	0.0125 *** (6.21)	0.0078 *** (4.12)	0.0258 (1.25)	0.0227 (0.80)
_cons	−12.2220 (−1.54)	−20.3087 ** (−2.02)	−29.4327 (−2.06)	29.5654 (1.56)
Control variables	Yes	Yes	Yes	Yes
Obs	8044	8875	8044	8875
Pseudo R^2^	0.1159	0.0999	0.0401	0.0459

Note: Except for the regression coefficient reported by _cons, the other independent variables in the table report the average marginal effect. The statistical values (Z value and *t* value, respectively) are in parentheses. *** and ** are significant at the level of 1% and 5% respectively.

## Data Availability

Data use of the 2017 China Household Finance Survey (CHFS) is supported and approved by China Household Finance Survey and Research Center at Southwest University of Finance and Economics in China. Official website: https://chfser.swufe.edu.cn/datas/ (accessed on 2 February 2022).

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
