# Peer review of "Family Life Cycle, Asset Portfolio, and Commercial Health Insurance Demand in China"

_ijerph, 2022, doi:10.3390/ijerph192416795_

Round 1
Reviewer 1 Report
1. Latest references are missing. The introduction must be improved.
2. The survey is based on the data collected from the China Household survey. Are the results obtained also valid for other countries?
3. The table data should be explained in detail.
4. The English Language should be improved.
5. Separate section of future scope should be included.
Reviewer 2 Report
The topic chosen for your article turns out to be interesting and, in essence, well addressed but, in my opinion, requires an additional little effort to make it truly appealing.
The writing, in my opinion, a general discourse should be made to make more understandable the research question, the analysis methodology used, and the results obtained. As it is these things exist but are addressed in points, they should be harmonized, that makes writing and reading more linear, fluid and less schematized. The methodological part should be better described, it is necessary to define why the Logit and Heckman model are then introduced, in this way the results obtained will be better understood. I suggest to add a discussions part, in which you compare your work with other authors, this it will also help to increase the number of references.
Here are some more specific suggestions:
Abstract: please added an initial sentence which introduces the topic of the proposed research.
Introduction: there is no bibliographic reference even though various information is given. Please introduce in the initial part the research topic and explained it. It is necessary to “generalize” the work a little to make it attractive and interesting even for a non-Chinese reader. I suggest to add a map of the study area to make it clear to an international reader which areas of China this work refers to.
Section 3.1 Variable selection and explanation: it could be a single part without subsection.
Please check the spaces (especially after the points), also check uppercase/lowercase (sometimes the letter is capitalized after the comma, for example, line 243: “this” no “This”), pay attention to repeated text parts (example: line 284: robust robust).
When entering numbers, if more than 3 digits, it is necessary to indicate thousands (example at line 211: 16919 should be 16,919).
line 514: better children?, please check the sentence;
line 203: stata should be Stata.
in the bibliographic references given in the text it is better to use “and” rather than “&”.
Round 2
Reviewer 1 Report
The comments are taken care of and updated in the paper.